# Focus on the Use of Resveratrol in Bladder Cancer

**DOI:** 10.3390/ijms24054562

**Published:** 2023-02-26

**Authors:** Alessandro Zucchi, Francesco Claps, Antonio Luigi Pastore, Alessandro Perotti, Andrea Biagini, Luana Sallicandro, Rosaria Gentile, Concetta Caglioti, Federico Palazzetti, Bernard Fioretti

**Affiliations:** 1Department of Translational Research and New Technologies, University of Pisa, 56126 Pisa, Italy; 2Department of Medicine, Surgery and Health Sciences, University of Trieste, 56126 Trieste, Italy; 3Department of Urology, ICOT Latina, Polo Pontino, La Sapienza University of Rome, 04100 Latina, Italy; 4Department of Medicine and Surgery, Perugia Medical School, University of Perugia, Piazza Lucio Severi 1, 06132 Perugia, Italy; 5Department of Chemistry, Biology and Biotechnologies, University of Perugia, Via dell’Elce di Sotto 8, 06132 Perugia, Italy; 6Department of Biology and Biotechnology, University of Pavia, 27100 Pavia, Italy

**Keywords:** apoptosis, bladder cancer, angiogenesis, proliferation, resveratrol

## Abstract

Bladder cancer is the most common tumor of the urinary system, with a high incidence in the male population. Surgery and intravesical instillations can eradicate it, although recurrences are very common, with possible progression. For this reason, adjuvant therapy should be considered in all patients. Resveratrol displays a biphasic dose response both in vitro and in vivo (intravesical application) with an antiproliferative effect at high concentrations and antiangiogenic action in vivo (intraperitoneal application) at a low concentration, suggesting a potential role for it in clinical management as an adjuvant to conventional therapy. In this review, we examine the standard therapeutical approach to bladder cancer and the preclinical studies that have investigated resveratrol in xenotransplantation models of bladder cancer. Molecular signals are also discussed, with a particular focus on the STAT3 pathway and angiogenic growth factor modulation.

## 1. Introduction

Resveratrol (trans-3,4′,5′-trihydroxystilbene) is a stilbene that consists of two aromatic rings joined together by an ethylene bridge. It is found in several plants and fruits, such as grapes (*Vitis vinifera*), mulberries (*Morus* spp.), and peanuts (*Arachis hypogaea*) [1]. Resveratrol is also a phytoalexin, a compound with an antibiotic function that is produced by higher plants in response to infections or other stressors [2]. Therefore, the concentration of resveratrol in plants increases in response to environmental stress, heavy metals, and UV light [3]. Resveratrol was isolated for the first time from the roots of *Veratrum grandiflorum* in 1940 and, subsequently, in 1963 from the roots of *Polygonum cuspidatum*, a plant used in traditional Chinese and Japanese medicine [2], rich in resveratrol 3-O-β-d-glucoside (piceid) [4].

Resveratrol has shown several beneficial properties for human health, such as anti-aging [5], anti-inflammatory [6], neuroprotective [7], hepatoprotective [8], cardioprotective [9], antidiabetic [10], and antioxidant activity [8]. Resveratrol displays chemopreventive action and anticancer action in several types of neoplasia [11]. Recently, resveratrol has attracted great interest for application in bladder cancer therapy [12]. The main obstacle in understanding the health potential of resveratrol is the difficulty of comparing the effects observed in vitro with those observed in vivo as the concentrations reached in the latter are lower than those that can be studied in vitro. With these limitations and the pharmacokinetics of resveratrol in mind, we critically reviewed in vivo studies (animal models) that have sought to evaluate the uses of resveratrol in the treatment of bladder cancer.

## 2. Bladder Cancer

Bladder cancer is the seventh most frequent cancer in the male population worldwide, with 9.5 cases per 100.000 people/year in men and 2.4 cases per 100.000 people/year in women. The incidence of bladder cancer in Europe is significantly higher than in the rest of the world, with 20 cases in men and 4.6 cases in women per 100.000 people/year [13]. The incidence in the world changes significantly in relation to the various risk factors to which the population is exposed, as well as due to the different diagnostic techniques and the availability of treatments [14]. At least 75% of the patients present, at the first diagnosis, non-muscle-invasive bladder cancer (NMBC) at different stages (Ta, carcinoma in situ, or T1), with a higher percentage in younger patients [15]. Smoking is the major risk factor in the pathogenesis of urothelial bladder cancer and the main causative agent in at least 50% of cases of bladder cancer. The risk increases progressively with the intensity and duration of smoke exposure.

Bladder cancer is classified using the TNM Classification of Malignant Tumours (TNM), approved by the Union for International Cancer Control (UICC). This classification is essential to define the appropriate treatment for each individual case of bladder cancer [16]. When the bladder tumor is removed endoscopically, it is possible to establish the “T” of the TNM, i.e., to define whether the tumor is non-muscle invasive (Ta and T1, which indicate whether it invades the lamina propria or not, respectively) or T2 (which indicates that it is muscle-invasive). The so-called “CIS” (carcinoma in situ), a flat, non-invasive, high-grade cancer, deserves special mention. It can be invisible to cystoscopy and, therefore, may not be diagnosed or may be interpreted as a simple inflammatory area due to its usually reddish appearance [17]. Therefore, the “T” of the TNM, together with the histology of the bladder tumor, is essential to define the therapeutic or follow-up procedure to adopt. Regarding histology, over 90% of bladder tumors are urothelial, and they are classified on the basis of the risk of progression (but not of recurrence) with a double classification: WHO 1973, which divides bladder tumors into G1, G2, or G3; WHO 2004/2016, which divides them into papillary urothelial neoplasm of low malignant potential (PUNLMP), low-grade, and high-grade [18,19]. Cancer is mostly non-muscle-invasive in the early stages, and up to 15% will eventually progress to muscle-invasive bladder urothelial carcinoma [20]. Superficial bladder cancers, such as stages Ta (superficial), Tis (in situ), and T1 (tumor invades subepithelial connective tissue) account for 75–85% of neoplasms at clinical presentation, while the remaining 15–25% are invasive (T2, T3, and T4) or have metastasized at the time of diagnosis [21].

As already mentioned above, the management of bladder cancer requires correct staging using the TNM system. The “T” of TNM is obtained by transurethral resection surgery. Transurethral resection of bladder tumor (TURBT) can stage up to T2, and it alone is capable of eradicating completely a Ta or T1 tumor. However, recurrences in bladder cancer are very common, with a possible progression. More than 70% of all patients treated for superficial bladder cancer will subsequently develop one or more recurrent tumors, and about one-third of these patients will progress to cancer that invades the surrounding muscle [22,23].

Adjuvant therapy should be considered in all patients (Figure 1): in low-grade bladder tumors, a single dose of intravesical chemotherapy (mitomycin C, epirubicin, or pyrarubicin) is suggested within 24 h of resection to prevent recurrence [24,25,26,27,28]. More intravesical instillations of chemotherapy may be necessary and suggested depending on the risk of progression and recurrence: for those patients with low-grade, low-risk tumors, only a single instillation post-surgical resection may be advised. Conversely, high-risk patients benefit from repeated treatment over time [29]. Patients with histological evidence of high-grade bladder cancer should undergo bladder instillations with bacillus Calmette–Guérin (BCG), which has a higher efficacy than chemotherapy in preventing the recurrence of non-muscle-invasive high-grade urothelial cancer [30,31,32,33,34]. CIS deserves a separate mention also regarding the treatment: in fact, it cannot be treated only with an endoscopic resection procedure. When a histological diagnosis of CIS is present, it is mandatory to perform intravesical instillations of BCG or, in certain cases, even to propose a radical cystectomy. There are some cases where the bladder tumor does not respond to BCG instillations and “relapsed tumors” or “recurrent tumors” are reported. In these cases, radical cystectomy must be proposed due to the high risk of progression and metastasis that BCG-unresponsive cancer has [35].

## 3. Resveratrol and Bladder Cancer

The in vitro and in vivo effects of resveratrol as an antineoplastic agent in bladder cancer have recently been reviewed [11,12]. While several researchers have studied the in vitro effects of resveratrol in bladder cancer biology, only two studies have considered its potential use in in vivo models. Notably, clinical studies on the use of resveratrol in bladder cancer therapy protocols are completely missing (evaluated in https://clinicaltrials.gov/ website by using keywords such as “cancer bladder” and “resveratrol”, accessed on 4 January 2023). In this review, we focus on the possible applications of resveratrol in bladder cancer treatment by considering the evidence derived mainly from in vivo models.

A resveratrol dose of 20 mg/kg body weight, with daily intraperitoneal (i.p.) administration, inhibited the growth of subcutaneous (s.c.) xenografted bladder cancer [36]. Other investigators used resveratrol i.p. administration to evaluate the efficacy of resveratrol to inhibit tumor cell growth different from bladder cancer. In s.c. xenografted ovarian cancer using Balb/c nu/nu mice, i.p. injection at concentrations of 50 and 100 mg/kg body weight for 4 weeks reduced tumor growth [37]. Similarly, concentrations of 20 and 40 mg/kg body weight reduced the s.c. growth of tumors from Erlich’s ascites [38]. In addition, in an s.c. neuroblastoma tumor model, 5 mg of resveratrol decreased cancer growth [39]. Surprisingly, 10 mg/kg body weight of i.p. resveratrol was not able to reduce cancer growth and survival in NOD.CB17-Prkdcscid/J mice engrafted with the human t(4;11) acute lymphoblastic leukemia (ALL) cell line [40].

The lack of anticancer effects of 10 mg/kg daily of i.p. resveratrol can be associated with the low dose used since, in other studies, the relationship between anticancer effects and dose has been observed. For example, in glioblastoma s.c. syngeneic rat xenotransplants models, the i.p. injection of resveratrol (10 or 40 mg/kg daily for 4 weeks) reduced tumor mass only at the highest concentration through an antiangiogenic action [41]. Interestingly, an antiangiogenic action of resveratrol (reduction of VEGF and FGF-2) was suggested to be involved in the tumor reduction of the s.c. xenotransplated human T24 bladder cancer model [36]. Overall, i.p. resveratrol displays anticancer effects in bladder tumor xenotransplantation similar to other neoplasia at an i.p. dose higher than 10 mg/kg.

A pharmacokinetic study of i.p. administration of 10 mg/kg of resveratrol in a single dose in mice displayed a sub-micromolar plasma concentration after 1 h from the administration (the serum concentrations of the total resveratrol were ~4 ± 2 μM, roughly distributed at a 1:3:1 ratio of resveratrol/resveratrol glucuronide/resveratrol sulfate [40]), indicating that this value represents the threshold to observe the cancer growth reduction in vivo [41]. Unfortunately, scant information was reported in in vitro studies of the effects of a low micromolar concentration of resveratrol on bladder cancer biology. This last consideration made it difficult to compare and use the data obtained from the in vitro experiment where resveratrol was tested in bladder cancer cell lines at concentrations higher than 10 μM and up to 200 μM [12].

Resveratrol showed a biphasic effect on proliferation, which was related to the concentration when it was applied to a bladder cancer cell line. Specifically, at low concentrations (lower or equal to 20 μM), it had no antiproliferative effects, while at high concentrations (greater or equal to 20 μM), it exhibited an antiproliferative effect by inducing apoptosis [42]. This outcome was confirmed in several studies, as in the bladder cancer cell line T24, BTT739, Pumc-91/ADM, where an antiproliferative effect (i.e., cell cycle blockade or apoptosis) was observed at concentrations higher than 20 μM [36,43,44,45,46,47,48]. In a few studies, the antiproliferative action was observed to be time-dependent [49], whereas, in others, it depended on the status of the tumor protein p53 (TP53) [47]. At low concentrations (2.5 μM), resveratrol did not show any antiproliferative activity but retained the ability to induce the mitochondrial BCL2 apoptosis regulator (BCL2) protein and BCL2-associated agonist of cell death (BAD) after 48 h of treatment, without a significant change in the BAD/BCL2 ratio [42]. Although the mechanism is still unclear, resveratrol reduced the growth of the tumor mass in vivo when treated via i.p. (low systemic concentration), by modulating processes such as angiogenesis [36], with a reduction of the vascular endothelial growth factor (VEGF) and fibroblast growth factor 2 (FGF-2). Therefore, we can hypothesize (Figure 2) that at low concentrations (obtained through a systemic administration, for example, i.p.) resveratrol has an antiangiogenic effect, reducing FGF-2 and VEGF, while at high concentrations (obtained through a local intravesical administration [44]) it has an antiproliferative effect as a consequence either of the reduction of miRNA21 [49], the signal transducer and activator of transcription 3 (STAT3), or its downstream genes, such as c-Myc, cyclinD1, survivin, and VEGF [44].

## 4. Other Mechanisms of Action of Resveratrol

The beneficial effects of resveratrol were associated with antioxidant properties and the ability to activate the sirtuins and protein kinase AMP-activated (AMPK) pathway. Because of the presence of more than one phenolic group, resveratrol belongs to the category of polyphenols and shows strong antioxidant properties, since it reacts with free radicals, resulting in more stable adducts, which are therefore less reactive and less toxic than the radicals themself [1], enhancing the cellular antioxidant activity. For example, resveratrol upregulates the tumor suppressor phosphatase and tensin homolog (PTEN), the major antagonist of phosphatidylinositol-4,5-bisphosphate 3-kinase (PI3K), by blocking AKT serine/threonine kinase 1 (Akt) activation, which leads to an upregulation of the mRNA levels of antioxidant enzymes such as catalase (CAT) and superoxide dismutase (SOD) [50]. Resveratrol stimulates the nuclear factor erythroid 2-related factor 2 (*Nrf2*), which initiates the transcription of many antioxidant genes such as *SOD* and *CAT* to reduce oxidative stress. Resveratrol could also improve the antioxidant defense system by modulating the action of antioxidant enzymes through the downregulation of the extracellular signal-regulated kinase (ERK) kinases that are activated by the reactive oxygen species (ROS). Neoplastic progression is associated with the alteration or mutation of genes that can occur spontaneously or following exposure to carcinogens. Oxidative stress plays a crucial role in the carcinogenesis process; ROS can react with DNA, causing serious damage, such as mutations [51]. Resveratrol, as a radical scavenger, appears as an anticancer agent by limiting the genotoxic impact of ROS and attenuating the processes of transformation into neoplastic cells [52]. In general, the anticancer effects have been correlated with other mechanisms of action independent from its radical scavenger properties [53]. In bladder cancer cell lines, resveratrol exerts its anticancer activity by inducing cell cycle arrest, apoptosis, differentiation, and inhibition of the proliferation of tumor cells.

It was demonstrated that resveratrol can activate the silent mating type information regulation 2 homolog 1 (SIRT1) and mimic the same beneficial effects induced by caloric restriction [54]. SIRT1 is a nicotinamide adenine dinucleotide (NAD)-dependent protein, one of the seven members of sirtuins, and belongs to the large family of the mammalian class III histone deacetylases [55]. It is mainly localized in the nucleus [56], and it is encoded by *Sir2* (silent information regulator 2) [57], a highly preserved gene; homologs of *Sir2* have also been found in lower organisms, such as the yeast *Saccharomyces cerevisiae*, the nematode *Caenorhabditis elegans*, and the dipterous *Drosophila melanogaster* [58,59]. The stimulation of SIRT1 leads to the deacetylation of lysine, coupled with the breakdown of NAD^+^ into nicotinamide adenine mononucleotide (NAM) and 1′-O-acetyl-ADP-ribose [59] or 1′- and 2′-O-acetyl-ADP-ribose [60], resulting in the modulation of the activity of the peroxisome proliferator-activated receptor gamma coactivator 1 alpha, (PPARGC1A, or PGC-1α) and other transcriptional factors related to aging and life span [47,61], including mitochondrial biogenesis [62].

However, even if the effects of resveratrol on SIRT1 and PGC-1α are well-known, the mechanism behind the regulation is still controversial [63]. Some studies have reported an indirect activation of SIRT1 by resveratrol, which firstly acts on AMPK, leading to an increase in NAD^+^ levels and, as a consequence, increases in SIRT1 and PGC-1α [64]. On the other hand, other studies have found a direct stimulation of SIRT1 by resveratrol, followed by the activation of AMPK through the deacetylation and activation of serine/threonine kinase 11 (LKB1) [65]. Data from in vitro and in vivo studies indicate a dose-dependent mechanism of resveratrol: when the dose of resveratrol was moderate (25 μM), the activation of AMPK was SIRT1-dependent, similar to that which occurs during caloric restriction, whereas a 2-fold concentration (50 μM) of resveratrol resulted in a SIRT1-independent activation of AMPK activation [63]. Furthermore, murine models lacking SIRT1 showed no differences in mitochondrial activity following treatment with both doses.

In osteoporosis rats, treatment with a high dose of resveratrol revealed a downregulation of Akt phosphorylation and a mechanistic target of rapamycin kinase (mTOR) phosphorylation, suggesting an involvement of the Akt/mTOR pathway in the bone cell autophagy activation induced by resveratrol [66] and the upregulation of the insulin signaling pathway through the phosphorylation of insulin receptor substrate 1 (IRS-1), PI3K, pyruvate dehydrogenase kinase 1 (PDK-1), Akt, and glycogen synthase kinase 3 (GSK-3) [67]. A double-blind randomized trial showed an improvement in the insulin sensitivity in T2D patients after 3 g/die of resveratrol per 12 weeks, with a significant increase in the SIRT1 and AMPK expressions in the skeletal muscle, which also determined an upregulation of the solute carrier family 2 member 4 (GLUT4) [68].

## 5. Pharmacokinetics of Resveratrol

Because of its low solubility in water, resveratrol shows a limited bioavailability which complicates the possibility of replicating in vivo what has been demonstrated by in vitro studies [47,69]. Once ingested through food or as a food supplement, resveratrol is rapidly assimilated in the small intestine [70,71] by passive diffusion [72] or by carrier [73]. In the enterocyte, it undergoes conjugation with uridine diphosphoglucuronic acid (UDP-GA) or 3′-phosphoadenosine-5′-phosphosulphate (PAPS), in reactions respectively mediated by different enzymatic isoforms of UDP-glucuronyltransferase (UGT) and cytosolic sulphotransferase (SULT) [74,75,76]. The conjugated resveratrol moves from the enterocyte through the transporters breast cancer resistance protein (BCRP) and multidrug resistance-associated protein 2 (MRP2) located on the apical membrane or through MRP3 on the basolateral side, entering the portal vein system until it reaches the liver. After that, the conjugation can further be catalyzed by UGT1A1, UGT9A, and SULT1A1—the same isoforms mainly expressed in the intestine [77,78]. From the hepatocyte, free resveratrol or its metabolites, including resveratrol-3-O-sulphate, resveratrol-3-O-4’-O-disulphate, and resveratrol-3-O-glucuronide, undergo enterohepatic recirculation and are re-absorbed in the small intestine before entering the portal system and reaching the liver. Here, they can go through new reactions or head into the systemic circulation to be distributed to the whole body and definitively eliminated through the urine (Figure 3). Alternatively, resveratrol can be absorbed in the large intestine, specifically in the colon, where it is metabolized by the intestinal microbiota with the formation of dehydro-resveratrol, lunularin, or 3,4’-dihydroxy-trans-stilbene [77], or eliminated through the feces [70]. Otherwise, similarly to what is described above, free resveratrol or its metabolites can reach the liver and undergo enterohepatic recirculation or enter the systemic circulation and reach the kidneys, to be excreted [77,78].

The pharmacokinetics of trans-resveratrol can change according to the type of administration, dosages, and protocol treatment [79], while its plasma concentration depends on the ingested dose [77]. Despite this, a study revealed a low plasma level concentration of trans-resveratrol following a high dose intake and a short dosing interval, with a circadian variation that was correlated with higher bioavailability after morning administration [80]. Resveratrol was well-absorbed when administered orally in 500 mg tablets, and its plasma concentrations or metabolites corresponded to the concentrations of its in vitro efficacy [81]. In addition, another study reported a plasma concentration of 1 μM of resveratrol and much higher concentrations of its glucuronide and sulfate conjugates [82].

Several approaches to improve resveratrol’s bioavailability and pharmacokinetics profile have provided promising results, such as nanocrystals [83], casein nanoparticles [84], liquid micellar formulations [85], self-emulsifying drug delivery systems [86], oat protein–shellac nanoparticles [87], and layer-by-layer nano-formulations [88]. Selective organ targeting is also possible with trans-resveratrol-loaded mixed micelles, which target the brain [89], and resveratrol-loaded glycyrrhizic acid-conjugated human serum albumin nanoparticles, which target the liver [90]. Recently, we developed a solid dispersion of resveratrol supported on magnesium dihydroxide (Resv@MDH) with better water solubility in simulated gastric fluids and an improved pharmacokinetic profile and bioavailability. Our investigation demonstrated that Resv@MDH increased resveratrol bioavailability by three times after oral administration in rabbit compared to pure resveratrol [91]. A clinical study also demonstrated that Resv@MDH improved the pharmacokinetic profile in humans [92], with a peak of plasma concentration in the micromolar range.

## 6. Combination of Resveratrol with Other Compounds

The use of resveratrol in combination with chemotherapeutic agents could allow for avoiding the development of drug resistance, which is a potential risk to not underestimate. Resveratrol can modulate and re-sensitize cancer cells to chemotherapeutic agents [37,93] when applied in combination with drugs in clinical therapy (Table 1).

The combination of resveratrol (75 and 150 μM) with gemcitabine (10 μM) in the T24-GCB cell line revealed an additive effect by reducing the cytoplasmic levels of deoxycytidine kinase (DCK), thymidine kinase 1 (TK1), and thymidine kinase 2 (TK2), while ATP binding cassette subfamily C member 2 (ABCC2) was increased [93,94]. Poly (ADP-ribose) polymerase (PARP) cleavage and apoptosis were also increased by the combined therapy. In addition, relatively low doses of resveratrol (10 μM) reduced the migratory ability of T24-GCB cells [94]. All these findings confirm the ability of resveratrol to reverse the drug resistance of T24-GCB cells to gemcitabine.

On the other hand, in vitro treatment with rapamycin (20 nM) and resveratrol (100 μM) in different cell lines (TSC1-null MEFS, WTMEFs, 639V, HCV29, and MGH-U1 cells) unveiled the efficacy of the combination in maintaining rapamycin-induced inhibition of mTOR and resveratrol-induced inhibition of Akt activation. In addition, the combined therapy of resveratrol and rapamycin upregulated PARP and caspase 3, inducing apoptosis and preventing cell migration and colony formation in TSC1-null MEFs but not in WTMEFs, suggesting a TSC1-dependent mechanism of action [95]. These data confirm the potential of combined therapy with resveratrol and rapamycin to inhibit bladder cancer cell growth and induce cancer cell death, which could be specifically fitted for bladder cancer patients with tumors characterized by TSC1 mutations or activating PI3K/mTORC1 pathway mutations.

Another study investigated the role of resveratrol (0, 10, 50, and 100 µM) on pumc-91/ADM cells, showing a decrease in the resistance to adriamycin but an increase in the cytotoxicity of the drug by upregulating the expression levels of DNA topoisomerase II (*TOP2*) and downregulating the expression levels of glutathione S-transferase (*GST*), LDL receptor-related protein (*LRP*), *BCL2*, and *MRP1* [45].

Combined therapy with doxorubicin at a low dose (2 µM) and resveratrol at high doses (150, 200, and 250 µM) in the 5637 and T24 bladder cancer cell lines showed an additive effect between the molecules, which caused enhanced cytotoxicity in both cell lines. Moreover, the combination of doxorubicin and resveratrol was more effective on the oxidative stress, cell colony formation, cell morphology, cell migration, and nuclear division index (NDI) assay [96] compared to the treatment with doxorubicin and resveratrol alone.

## 7. Final Remarks and Perspectives

Resveratrol displays potential anticancer activity in vivo, but data for synergetic effects with anticancer agents are missing. In glioblastoma, the combined action of resveratrol with anticancer agents needs pre-incubation [97]. In bladder cancer, this has not yet been studied, but it should be considered in order to be able to schedule the administration times in association with other chemo and radio-therapeutic agents. Another aspect of the future of the use of resveratrol for bladder cancer is the development of new formulations with better bioavailability and plasma concentrations. In fact, resveratrol has a plasma peak and low bioavailability that limit its efficacy as an anticancer agent since the plasma levels necessary for therapeutical effects are difficult to reach with common formulations [98,99]. Thus, research on a better resveratrol formulation with an increased pharmacokinetic profile represents a challenge for its potential use as adjuvant therapy in bladder cancer.

The outcomes from the combination therapy of resveratrol and drugs such as gemcitabine and rapamycin suggest a key role of resveratrol as an adjuvant in clinical therapy. The high concentrations (from 75 μM to 250 μM) indicated in the studies may be obtained using intravesical administration. However, the potential cytotoxic effect should be considered not only on the cancer cells but for the whole surrounding environment too. Therefore, we propose the instillation of resveratrol through intravesical injection after or before treatment, especially with gemcitabine, adriamycin, and doxorubicin, as rapamycin is cytostatic and not cytotoxic, in order to reduce the cytotoxicity of the combination.

Another crucial aspect regarding resveratrol is its action on miRNA21. As already highlighted, resveratrol can downregulate miRNA21 expression, especially when high doses are injected through intravesical administration. The expression of miRNA21 is high in bladder cancer and stromal cells and strictly related to cancer development [100], as it increases cancer progression by polarizing tumor-associated macrophages (TAMs) [101]. These macrophages are similar to M2 macrophage phenotypes and inhibit the function of cytotoxic T-lymphocytes (CTL). Moreover, the inhibition of miRNA21 was also correlated with the suppression of the Warburg effect in the osteosarcoma MG-63 cell line, through the reduction of the levels of lactic acid, adenosine triphosphate (ATP), and glucose uptake, and the downregulation of those proteins involved in the Warburg effect, such as GLUT1, lactate dehydrogenase A (LDHA), hexokinase 2 (HK2), and pyruvate kinase M1/2 (PKM) [102]. A similar effect is also plausible in bladder cancer, as indicated by the literature available so far.

Further studies should be carried out to better understand how these molecules work together and enhance the effectiveness of the standard clinical approach. These results could pave the way for new opportunities, such as the use of this novel resveratrol formulation in clinical applications as an adjuvant therapy for bladder cancer.

## Figures and Tables

**Figure 1 ijms-24-04562-f001:**
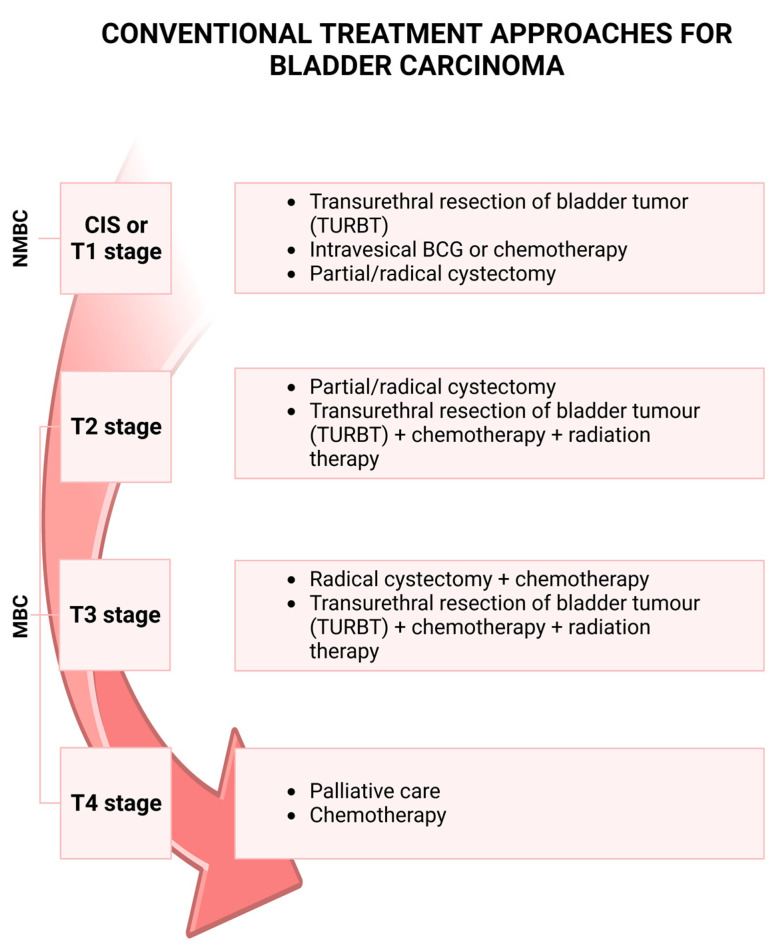
Bladder carcinoma progression and therapy. Non-muscle-invasive bladder cancer (NMBC), including the carcinoma in situ (CIS) or T1 stages, is treated with transurethral resection of bladder tumor (TURBT), intravesical instillation of bacillus Calmette–Guérin (BCG) or chemotherapy. Treatment for muscle-invasive bladder cancer (MBC) can be cystectomy with chemotherapy or TURBT with chemotherapy and radiation therapy. The conventional treatment for the T4 stage is chemotherapy or palliative care. Created with BioRender.com.

**Figure 2 ijms-24-04562-f002:**
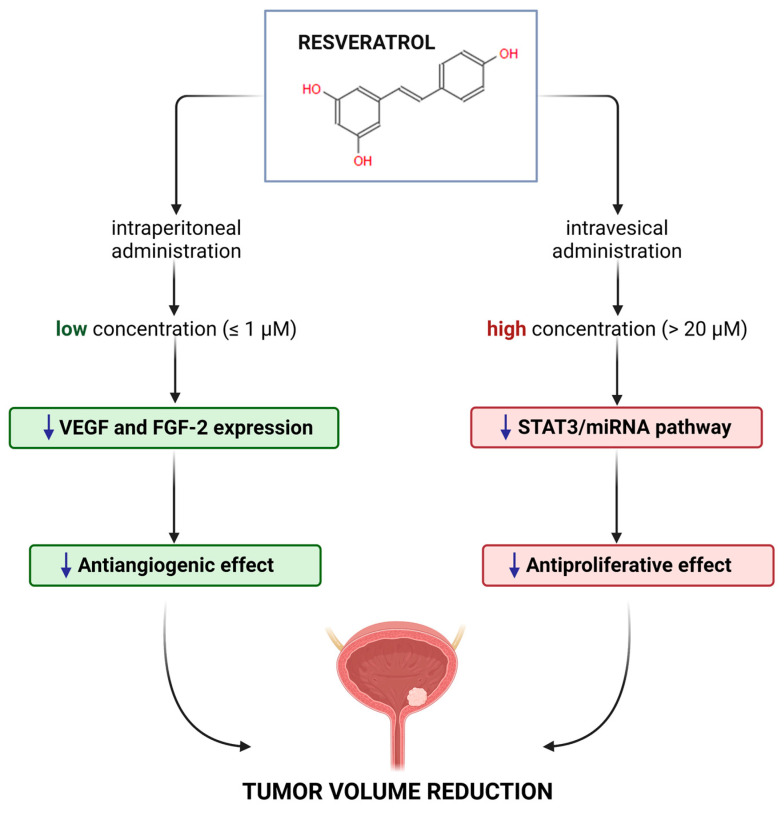
Effects of resveratrol on bladder cancer in in vivo models. At low concentrations (achievable through an intraperitoneal administration, i.p., lower or equal to about 1 μM [40]), resveratrol exhibits an antiangiogenic effect, reducing vascular endothelial growth factor (VEGF) and fibroblast growth factor 2 (FGF-2). On the other hand, at high concentrations (achievable through an intravesical administration, i.v., higher than 200 μM [44]), it shows an antiproliferative effect, reducing miRNA21, the signal transducer and activator of transcription 3 (STAT3) and its downstream genes. Created with BioRender.com.

**Figure 3 ijms-24-04562-f003:**
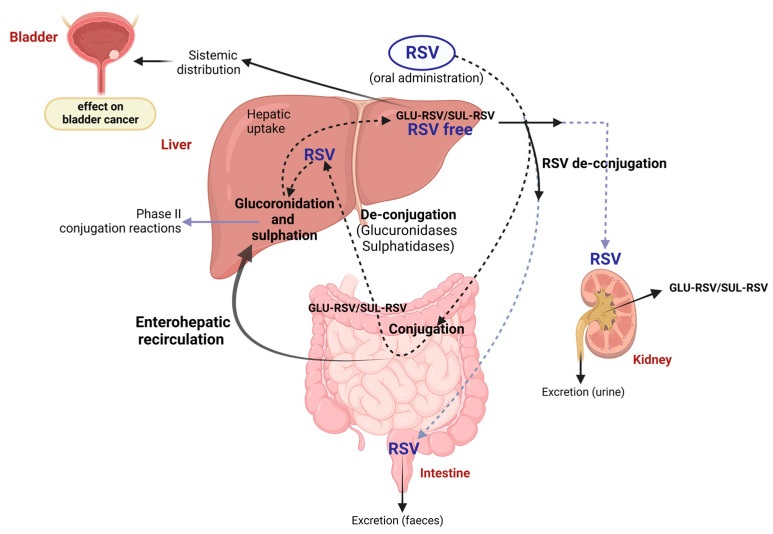
Metabolism and biotransformation of resveratrol in different organs. Resveratrol undergoes conjugation with uridine diphosphoglucuronic acid (UDP-GA, indicated as GLU-RSV in the figure) or 3′-phosphoadenosine-5′-phosphosulphate (PAPS, SUL-RSV indicated as SUL-RSV in the figure) in the enterocytes, before reaching the liver through the portal bloodstream. Here, it can be further conjugated (phase II reactions) and then absorbed by peripheral tissues. Part of the resveratrol intake can reach the plasma as free resveratrol (RES free), and another part can come from the de-conjugation reaction. In addition, conjugated resveratrol and its metabolites undergo enterohepatic circulation, where it can be metabolized by the intestinal microbiota. In the end, its excretion can occur through urine or feces. Created with BioRender.com.

**Table 1 ijms-24-04562-t001:** Effects of treatment with resveratrol in combination with different drugs in bladder cancer cell lines. The table describes the effects of the combination between resveratrol and other compounds in the treatment of bladder cancer. DCK = deoxycytidine kinase; TK1 = thymidine kinase 1; TK2 = thymidine kinase 2; ABCC2 = ATP binding cassette subfamily C member 2; PARP = poly (ADP-ribose) polymerase; mTOR = rapamycin kinase; Akt = AKT serine/threonine kinase 1; TOP2 = DNA topoisomerase II; GST = glutathione S-transferase; LRP = LDL receptor-related protein; BCL2 = BCL2 apoptosis regulator; MRP1 = multidrug resistance-associated protein 1.

Combination	Cell Line	Effect	Reference
**Resveratrol + Gemcitabine**	T24-GCB	Decreased DCK, TK1, TK2. Increased ABCC2, PARP, apoptosis. Reduced cell migration.	[94]
**Resveratrol +** **Rapamycin**	TSC1-null MEFs WTMEFs 639V HCV29 MGH-U1	Inhibition of mTOR and Akt. Upregulation of PARP and caspase 3 (apoptosis). Inhibition of cell migration.	[95]
**Resveratrol + Adriamycin**	Pumc-91	Decreased resistance. Increased cytotoxicity. Upregulation of TOP2. Downregulation of GST, LRP, BCL2, MRP1.	[45]
**Resveratrol +** **Doxorubicin**	5637 T24	Increased cytotoxicity. Reduced cell migration and colony formation. Reduced oxidative stress.	[96]

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
