# Peer review of "Focus on the Use of Resveratrol in Bladder Cancer"

_ijms, 2023, doi:10.3390/ijms24054562_

Round 1
Reviewer 1 Report
The manuscript of the article "Focus on the use of Resveratrol in bladder cancer" reviews current knowledge of the Resveratrol impact as a single agent as well as in combination with conventional chemotherapy agents in the treatment of cancer, with particular emphasis on bladder cancer.
In this review recent advances are discussed and graphically summarized in order to give a concise update to the reader. Overall, the manuscript is scientifically interesting and plausible, with up-to date literature citations.
However, throughout the text, style and grammar should be checked and improvements made.
For example: "Conclusion and prospective (or PERSPECTIVE) Resveratrol displays potential anticancer activity in vivo, but (DATA for) synergetic effects with anticancer agents are missing " - word DATA should be added in order to improve clarity of the sentence.
Author Response
We thank the reviewer for the suggestion. Style and grammar in the text have been improved, included the example reported by the referee.
Reviewer 2 Report
In this review article, the authors discussed the standard therapeutical approach to bladder cancer and the preclinical studies that investigated resveratrol in xenotransplantation models of bladder cancer.
Comments
The reviewer has some concerns as follows:
1. The sectioning in this article is very unclear and not easy to read. It should be revised.
2. The format of this manuscript does not meet the journal requirements. It should be revised.
3. The purpose of this review article should be stated in the Introduction.
4. The font size of the text within figure 2 can be bigger.
5. In Figure 2, both “low” and “high” concentration ranges (or dose ranges) of resveratrol can be shown.
6. In page 7, line 5, the sentence “Pharmacokinetic study of i.p. administration of 10 mg/kg in a single dose in mice…” is incomplete that resveratrol is lacking.
7. A summary for in vitro and in vivo effects of resveratrol on bladder cancer can be listed in a Table to facilitate the reader's comprehension.
Author Response
We thank the reviewer for the suggestion. Below, we give our replies to referee’s comments
- The new version of the article is revised according to the review
- In this new version, the article is revised according to meet the journal requirements
- In the new version of the Ms, we add a paragraph in the introduction to focalize the target of the review
- In the new version, the font size in figure was modified in agreement of the reviewer’s comment
- In the new version of the Figure, the new information about both “low” and “high” concentration is specified
- In the new version of the article, resveratrol is specified
- In this new version, we added a new table that summarized the articles that report the adjuvant on action of resveratrol with other drug according with focus of review.
Reviewer 3 Report
Resveratrol is a phenolic compound of plant origin that exerts pleiotropic, i.e. multidirectional, effects in the human body. Its effectiveness and safety have been documented in hundreds of scientific studies. The authors did not describe what resveratrol is and what health benefits may result from its use? Civilization diseases such as obesity, diabetes, cardiovascular diseases and cancer are a huge, growing problem faced by humanity. We are constantly looking for new drugs and dietary supplements that would have a positive effect on the course of these diseases and prevent their development. Natural preparations based on plant extracts are very popular. The authors did not present it either. The authors very modestly presented their research and conclusions in the manuscript. It looks very bad. Many scientific studies focus on resveratrol's health benefits. After all, the protective and healing effects of resveratrol in the course of diabetes, obesity, cancer, cardiovascular diseases, as well as some bacterial and viral infections have been analyzed. In people with type II diabetes, taking resveratrol at a dose of 1g/day for 45 days contributed to a significant reduction in fasting glucose, insulin concentration and insulin resistance. Used for 3 months, it reduced the concentration of glycated hemoglobin and total cholesterol, and also contributed to the regulation of hypertension. In a group of oncological patients, resveratrol had a positive effect on the level of some cytokines and compounds involved in the metastasis process and the development of inflammation. Patients with cardiovascular disease who were given resveratrol for 6-12 months saw a decrease in total cholesterol and apolipoproteins involved in the development of atherosclerosis. In addition to improving the lipid profile, there was a decrease in C-reactive protein (an acute phase inflammatory protein), fasting glucose and blood pressure. By reducing the level of tumor necrosis factor α (TNF-α), interleukin 6 (IL-6) and C-reactive protein, resveratrol may reduce the formation of inflammatory diseases. In addition, its beneficial effects can also be observed by women struggling with persistent menopausal ailments, because it affects the metabolism of estrogens, and also improves mood and memory in this group of people. The authors don't even mention it. An article in this form can not be accepted.
Author Response
We thank the reviewer for the suggestions, in the new version of the Ms we have added a paragraph about an overview of the general benefit effects of resveratrol, before starting to focus on the applications in bladder cancer therapy.
Reviewer 4 Report
Preparing a good review article on resveratrol is quite a challenge these days. Resveratrol is still a very intensively exploited topic. Therefore, it is challenging to prepare a manuscript that will bring something new, attract readers' attention, and thus deserve publication. This article does not meet these requirements. The authors undertook to reproduce the topic from the article DOI:10.1590/1678-4685-GMB-2020-0371, which they cited themselves. Unfortunately, nothing can say that this article brings any significant new content.
Moreover, the information contained in this article is much smaller than the work prepared by Almeida and Da Silva. The authors should see in this paper what the reader expects from a review titled “resveratrol and bladder cancer”. Almeida and Da Silva have prepared a comprehensive review of publications with specific data from in vivo and in vitro experiments summarized in clear tables. The paper reviewed here contains too long and too detailed histopathological characteristics of bladder cancers. It is unnecessary here and has almost no connection with the data presented in the next paragraphs where, the authors presented some randomly selected, commonly known, information about the anticancer effects of resveratrol, and some information concerning its pharmacokinetics and bioavailability is also provided. These data however are commonly known and therefore not interesting even for less experienced researchers. Specific data on resveratrol activity against bladder cancer cells are very laconic and trivial, often not supported by either the name of the cell line or references. Therefore I suggest rejecting this manuscript.
Author Response
We thank the reviewer for the suggestions. In this new version, we removed some aspects about the histopathological characteristics of bladder cancers. We added a new table to make it more systematic and we would like to clarify that our work completes Almeida's vision by describing the connection between pharmacokinetics profile of resveratrol with the beneficial effect in anticancer properties in bladder cancer. To this aim, we added a specific sentence in the introduction to highlight the originality of the paper “ The main obstacle in understanding the health potential of resveratrol is the difficulty of comparing the effects observed in vitro with those observed in vivo as the concentrations reached in the latter are lower than those that can be studied in vitro. With these limitations and the pharmacokinetics of resveratrol in mind we critically review in vivo studies (animal models) that seek to evaluate the uses of resveratrol in the treatment of bladder cancer”.
Reviewer 5 Report
General comments:
This review examines the standard therapeutical approach to bladder cancer and the preclinical studies that investigated resveratrol in xenotransplantation models of bladder cancer.
Major comments:
1. A table for “Combination of resveratrol with other compounds” is suggested to provide.
2. Subtitles of some paragraph are suggested to provided. Many paragraphs without subtitles sometime confuse the readers.
Minor comments:
1. The structure of resveratrol is suggested to provided.
Author Response
We thank the reviewer for the suggestion. Below, we give our replies to referee’s comments
Major comments:
- In this new version, a new Table is added to highlight the focus on the “Combination of resveratrol with other compounds”
- This new version has been revised according to the reviewer’s comment.
Minor comments
- The structure of resveratrol is added in Figure 2.
Round 2
Reviewer 2 Report
This revised manuscript can be accepted. No further comments.
Reviewer 3 Report
In addition, the world literature is very extensive about the presented research topic.
Reviewer 4 Report
The Authors have introduced suggested changes and significantly improved their work.